# Time to Rethink Bronchiolitis Obliterans Syndrome Following Lung or Hematopoietic Cell Transplantation in Pediatric Patients

**DOI:** 10.3390/cancers16213715

**Published:** 2024-11-04

**Authors:** Tang-Her Jaing, Yi-Lun Wang, Chia-Chi Chiu

**Affiliations:** 1Division of Hematology and Oncology, Department of Pediatrics, Chang Gung Memorial Hospital, 5 Fu-Shin Street, Kwei-Shan, Taoyuan 33315, Taiwan; g987669@gmail.com; 2Division of Nursing, Chang Gung Memorial Hospital, 5 Fu-Shin Street, Kwei-Shan, Taoyuan 33315, Taiwan; chi0105@cgmh.org.tw

**Keywords:** bronchiolitis obliterans syndrome, hematopoietic cell transplantation, lung transplantation, graft-versus-host disease, chronic lung allograft dysfunction

## Abstract

Bronchitis obliterans syndrome (BOS) may occur following lung transplantation (LTx) or hematopoietic cell transplantation (HCT). The primary issue is graft-versus-host disease, complicating diagnosis. Treatment is based on empirical evidence and interdisciplinary knowledge. Recent advances emphasize understanding the etiology, clinical features, and pathobiology of BOS, fostering cross-disciplinary knowledge. Treatment algorithms are based on thorough research and expert clinical insights, with new therapies being explored to enhance survival rates and prevent LTx or re-transplantation.

## 1. Introduction

### 1.1. Bronchiolitis Obliterans Overview

Bronchiolitis obliterans (BO) is a chronic lung condition causing small airway obstruction. It can be categorized into postinfectious BO, post-lung transplantation (LTx), and post-hematopoietic cell transplantation. Diagnosis relies on a thorough history, clinical evaluations, and radiological evidence, with lung biopsy and histopathology serving as the definitive standard. Lung biopsy may not be a viable option for every patient. A comprehensive strategy and expert attention are essential for effective treatment.

Bronchiolitis obliterans syndrome (BOS) after HCT is a relatively uncommon occurrence, affecting only a small percentage of recipients within five years [1]. The most prevalent noninfectious pulmonary complication following HCT is BOS, which represents a form of chronic lung graft-versus-host disease (GVHD) impacting 4.5–8.3% of children after HCT [2,3]. BOS is also a frequently observed form of chronic lung allograft dysfunction (CLAD-BOS) following LTx. Early detection of CLAD-BOS may be more feasible compared to BOS related to GVHD (GVHD-BOS) due to more frequent pulmonary function monitoring in lung transplant recipients. It affects approximately 50% of recipients within five years after the lung transplant and is a significant contributor to mortality in the later stages of post-transplantation [4]. BOS has a detrimental effect on the prognosis following LTx and HCT.

Postinfectious BO (PIBO) has been challenging to study due to its sporadic appearance and low incidence. The absence of systematic case registration and national or international databases on PIBO makes its incidence unpredictable [5]. Study-based protocols exist for evaluating and potentially treating CLAD-BOS and GVHD-BOS, but the diagnosis and management of PIBO remain less defined. This review provides an overview of the two types of BOS in children, emphasizing recent advancements after LTx and HCT, along with expert opinions, a literature review, and insights into new treatment strategies. This study aimed to perform a thorough review of BOS through an analysis of existing literature. 

### 1.2. What Are CLAD-BOS and GVHD-BOS?

#### 1.2.1. What Does CLAD-BOS Refer To?

The International Society for Heart and Lung Transplantation (ISHLT) characterizes CLAD as a consistent decline in pulmonary function. This decline is marked by a reduction of at least 20% in forced expiratory volume in 1 s (FEV1) for over 3 months from the baseline following transplantation. This definition is arrived at by excluding other potential causes (Table 1) [6]. The baseline value is determined by averaging the two most accurate post-operative FEV1 measurements taken at least three weeks apart. Staging the current FEV1 relative to the baseline determines the severity of CLAD.

CLAD clinical phenotypes are categorized by the main ventilatory pattern, total lung capacity (TLC), and whether opacities are present on chest CT scans [8]. Restrictive allograft syndrome (RAS) is characterized by pulmonary function changes such as decreased total lung capacity (TLC) and a FEV1 to forced vital capacity (FVC) ratio exceeding 0.7. On the other hand, BOS is characterized by stable or increasing TLC and a decline in FEV1/FVC ratio, indicating obstruction and hyperinflation [9]. Diagnosing RAS requires observing persistent opacities in multiple lobes of the parenchyma and/or pleura on a CT scan or a chest radiograph if CT is unavailable [6].

A majority of CLAD patients, around 65–70%, exhibit the BOS phenotype, while a smaller percentage, approximately 10–35%, display the RAS phenotype [10]. A recent study adhering to the latest ISHLT consensus criteria found a lower proportion of cases with RAS [4]. This change is due to the recognition of the mixed phenotype as a distinct entity, which was previously classified with RAS. Levy and colleagues estimate that about 5% of patients exhibit mixed phenotypes, while roughly 10% have undefined phenotypes [11]. 

Phenotypes can alter over time, with patients shifting from BOS to RAS and vice versa [4]. This transition is often referred to as BOS-to-RAS [12]. The survival rate for patients with BOS-to-RAS is lower than those with BOS but similar to or higher than those with RAS [13]. Factors contributing to both BOS and RAS encompass non-adherence to immunosuppressive regimens, episodes of acute cellular rejection (ACR), lymphocytic bronchiolitis, community-acquired respiratory viral infections, the presence of donor-specific antibodies, air pollution, gastroesophageal reflux disease (GERD), colonization by *Pseudomonas aeruginosa*, and CMV mismatch [4,14].

#### 1.2.2. What Is GVHD-BOS

The National Institutes of Health (NIH) criteria for diagnosing BOS in children post-HCT have limitations, leading to late diagnosis. Limitations include outdated pulmonary function tests (PFT), dependence on spirometry, and a fixed FEV1 threshold. Pulmonary complications occur in 25% to 60% of children after HCT. GVHD-BOS usually develops between 100 days and 2 years of HCT, but onset beyond 5–6 years has been noted [4]. Factors that contribute to the risk of GVHD-BOS encompass inadequate lung function, the use of a myeloablative/busulfan conditioning regimen, CMV seropositivity, a prior history of pulmonary disease before transplantation, a female donor, an unrelated donor, and a history of previous acute GVHD. Clinical biomarkers are necessary 80–100 days post-HCT to identify patients at risk [15]. Forced expiratory flow from 25% to 75% of the maximum is a key biomarker for early detection.

GVHD-BOS is a systemic condition affecting various organ systems, including the lungs, skin, nails, eyes, mouth, hair, genitals, joints, liver, and hematopoietic systems. Symptoms may consist of a continual cough or difficulty breathing during physical activity and can be identified through regular pulmonary function assessments [4]. The diagnosis resembles CLAD, characterized by primary impairment in pulmonary function. Nonspecific symptomatology may cause diagnostic delays.

## 2. Pathophysiology and Etiology of BOS

### 2.1. Immunology and Pathobiology of CLAD-BOS 

The development of CLAD-BOS in lung transplant patients is shaped by a range of mechanisms. These include alloimmunity, humoral immunity, autoimmunity, and factors such as lipopolysaccharide and toll-like receptors. Additionally, mediators like matrix metalloproteinase, interleukin, neutrophils, hyaluronan, and transforming growth factors play significant roles in this process [16]. Microbes and viruses contribute to BOS, with NF-κB activation blocking lung injury during allograft rejection [17]. High mobility group box protein 1 worsens fibrosis and contributes to BOS [18]. Cysteinyl leukotrienes, including LT B4, may play a role in BOS development [19,20]. Exposure to LT C4 leads to heightened expression of collagen I and smooth muscle actin, playing a role in the epithelial-to-mesenchymal transition associated with BOS development.

### 2.2. Immunology and Pathobiology of GVHD-BOS

The development of BOS in GVHD following HCT is not well understood [2]. Chronic GVHD shows autoimmune and fibrotic characteristics, whereas acute GVHD mainly consists of inflammatory elements [21]. The immune pathophysiology involves donor T-cells, B cells, auto-antibody production, and other donor cells, all of which are crucial components. The feedback loop with systemic and neutrophil autocrine IL-8 production leads to an activated, prothrombotic neutrophil phenotype characterized by degranulation and the creation of neutrophil extracellular traps [22]. GVHD patients have high levels of B-cell activating factor in their plasma. Fibrosis is a key feature of GVHD, primarily driven by TGF-β, TNF-α, and platelet-derived growth factors [23]. Higher levels of CD4+ and CD8+ T cells are thought to play a role in the development of BOS. CysLTs, macrophages, and neutrophils are present in the bronchoalveolar lavage (BAL) of these patients. Expression of leukotriene receptors is increased in macrophages, monocytes, eosinophils, granulocytes, and T cells [17,24]. Figure 1 illustrates the mechanisms and treatment associated with the development of BOS in the LTx or HCT.

## 3. Diagnosis

Early diagnosis of CLAD-BOS may be more likely in LTx recipients than GVHD-BOS patients due to more frequent monitoring of pulmonary function [4,25]. However, after HCT, PFT is less frequent, potentially missing opportunities for early intervention. Current guidelines recommend early investigations for CLAD, with spirometry being a key tool for monitoring and diagnosing pulmonary complications [26]. Home spirometry may detect early decline, but it requires calibration and patient assessment. Home spirometry may help limit clinic visits during the pandemic [27].

Lung function decline can be diagnosed using various diagnostic methods, including bronchoscopy, transbronchial biopsies, BAL, TLC testing, and CT imaging [28]. Lung biopsy is considered the gold standard, but risks may outweigh its benefits, especially in patients with GVHD-BOS [29]. Small airway brushings can reveal a lymphocytic gene expression signature in CLAD patients, which may be missed in transbronchial biopsies [30]. Functional magnetic resonance imaging (MRI) can evaluate regional lung function changes, but it is costly and not widely accessible [31]. CT is favored for its superior sensitivity and specificity, along with enhanced visualization of lung parenchyma and small airways. Air trapping may restrict the effectiveness of CT in patients with GVHD-BOS [4,32].

Chest CT findings in CLAD (RAS/mixed phenotype) include opacities and increased pleural thickening, which may be present in patients with BOS after LTx or GVHD [4,12]. Parametric response mapping (PRM) is an emerging modality for BOS diagnosis and assessment, correlated with FEV1 decline in patients without restrictive patterns. PRM detects BOS even with concurrent infection [33]. MRI has been studied for the morphological evaluation of transplanted lungs, with certain parameters possibly serving as early indicators of CLAD [34]. Artificial Intelligence (AI) has shown promise in enhancing our comprehension of rare diseases like lung GVHD post-HCT through quantitative imaging [4]. By leveraging AI, there is potential to deepen our insights into the similarities and distinctions between CLAD-BOS and GVHD-BOS [35].

Recent research emphasizes the significance of early diagnosis in enhancing patient outcomes. New CT strain metrics allow for earlier detection of BOS, even before significant declines in traditional measures such as FEV1 [36]. This advancement is vital as BOS can appear as early as three months after transplantation, resulting in progressive obstructive lung disease that particularly affects peripheral airways [37]. 

Revisions to the NIH criteria in 2020 have led to cohort studies that improve our understanding of the clinical presentation of post-HCT BOS [38]. The approach to BOS in other clinical settings, including LTx, has evolved. Potential solutions for the previously discussed issues are summarized in Table 2. 

## 4. Current Treatment Approaches

### 4.1. CLAD-BOS

Identify and treat underlying conditions like ACR, infection, lymphocytic bronchiolitis, and GERD. Optimize the patient’s immunosuppressive regimen. Early management of BOS can stabilize declining lung function. Avoid high-dose steroids and follow standard protocols. Switching from cyclosporine to tacrolimus may lower BOS prevalence after LTx [39,40,41].

Azithromycin is the most effective treatment for BOS, with a randomized, placebo-controlled trial showing a predicted FEV1 increase of 16–18% [42,43,44]. However, response rates were only 29–50% [45], with factors like airway neutrophilia and early treatment initiation contributing to better response [46]. Azithromycin is generally well tolerated, with gastrointestinal disorders being the most common adverse effects. The ideal dosage and duration of azithromycin for BOS remain undetermined [47]. Research indicates that azithromycin, used for short to mid-term periods (≤24 weeks), can be effective and safe for treating BOS [48]. However, optimal treatment options for PIBO have not been determined [49]. Azithromycin has been studied for its therapeutic effects on BOS after LTx [50]. Additionally, azithromycin has been explored in the treatment of lymphocytic airway inflammation, a risk factor for chronic lung allograft dysfunction [51].

Montelukast is an orally dosed drug that targets the CysLT receptors in the airways, specifically antagonizing LT D4 at the CysLT1 receptor. By binding to these receptors, Montelukast helps prevent smooth muscle contraction, reduces mucus secretion, and decreases airway edema. It is known for its selectivity for the CysLT1 receptor, making it an effective leukotriene receptor antagonist [52]. This is utilized in the management of chronic asthma, allergic rhinitis, exercise-induced bronchoconstriction, and neurodegenerative diseases. However, it has been linked to neuropsychiatric effects like depression, anxiety, sleep problems, agitation, and paresthesia [53]. These side effects tend to occur more frequently in individuals, both children and adults, who have a background of psychiatric disorders. Montelukast also exerts anti-inflammatory effects through innate immune cells and non-CysLT-dependent pathways, such as inhibiting NF-kB signaling and increasing antioxidants [54]. A pilot study showed that it slowed the FEV1 decline in patients with low neutrophilia on BAL and attenuated the FEV1 decline in 153 BOS patients. However, a randomized trial failed to show any effect on BOS patients [55]. 

Second-line treatments for CLAD-BOS encompass extracorporeal photopheresis (ECP) and total lymphoid irradiation (TLI) [4,6,56]. A European multicenter analysis highlighted that a significant portion of patients undergoing ECP had BOS, with a minority having RAS at the time of ECP initiation [57]. However, it is costly, not widely accessible, and may impose challenges. TLI slows FEV1 decline in BOS patients, but studies are small and observational [4,58]. Re-transplantation is an option for selected patients who do not respond to first- and second-line treatments. For those facing refractory allograft dysfunction, lung re-transplantation stands as the primary therapeutic avenue [59].

### 4.2. GVHD-BOS

Chronic GVHD or noninfectious complications, such as interstitial lung diseases, differ from BOS treatment [60]. The first step is to manage comorbidities and potential precipitating factors, optimizing immunosuppressive treatment. BOS treatment often complements the targeting of extrapulmonary GVHD manifestations, and long-term corticosteroids may be included in the therapy [4]. The European Society for Blood and Marrow Transplantation recommends a steroid pulse followed by a quick taper over one month, using a combination of fluticasone, azithromycin, and montelukast (FAM) [61]. Standard treatment for persistent GVHD was administered to the majority of patients, which helped manage BOS. The use of FAM is supported by preliminary data suggesting positive outcomes from individual components, particularly Montelukast, in treating BOS after HCT [62].

The data on azithromycin alone in treating BOS are less robust, with some studies suggesting relapse risk. Additionally, there are concerns regarding the long-term effects of azithromycin, particularly its association with an increased risk of developing second cancers in patients with BOS, as highlighted in several studies [63]. A combination of inhaled bronchodilators and corticosteroids has shown improvement in FEV1 in patients without systemic corticosteroids [64]. Supportive care, prophylaxis, pulmonary rehabilitation, nutritional support, and GERD treatment may also be beneficial [65]. Patients with chronic lung GVHD should be included in clinical trials with a focus on second-line treatment since there are currently no established methods for this [66]. Other agents like Bruton’s tyrosine kinase inhibitor ibrutinib and Janus kinase (JAK) inhibitor ruxolitinib are being explored [67,68,69]. In patients with post-transplant BOS, the short-term to mid-term (≤24 weeks) treatment with azithromycin significantly improves lung function, according to the overall results [48]. However, this improvement does not persist over time, and in fact, there is a worsening tendency in the long run. At the same time, azithromycin was well tolerated and demonstrated some survival improvements [70].

Azithromycin and ruxolitinib both affect fibroblast proliferation, though they operate via distinct mechanisms. One possible function of azithromycin is to modulate fibroblast activity in inflammatory circumstances, as it suppresses gingival fibroblast proliferation and activates matrix metalloproteinase-1 [71]. This antibiotic targets senescent cells and induces autophagy, potentially reversing certain fibrotic processes [72]. Azithromycin inhibits quorum sensing, reduces biofilm formation, and may have additional immune system effects, inhibiting proinflammatory cytokine production and potentially contributing to its therapeutic effects in certain inflammatory conditions [73]. In summary, azithromycin directly impacts fibroblast proliferation and autophagy, whereas ruxolitinib affects fibroblast activity by modulating cytokine signaling associated with inflammation. Both agents highlight different therapeutic strategies for treating conditions associated with fibroblast dysregulation, as shown in Figure 1. Ruxolitinib has been linked to a higher risk of infections [74], particularly herpes zoster, compared to other treatments like azithromycin. In contrast, azithromycin, known for its immunomodulatory effects, has shown potential antiviral benefits, especially in COVID-19. Therefore, ruxolitinib may predispose patients to viral infections [75]. Ruxolitinib is primarily used to treat myelofibrosis and other hematological malignancies. It inhibits cytokine signaling pathways essential for cell proliferation and survival, indirectly affecting fibroblast behavior through its influence on the inflammatory microenvironment [76].

## 5. New Perspectives and Next Directions

### 5.1. Hypothesis-Driven Novel Approaches

BOS management relies on understanding the disease pathogenesis and immunogenicity of the allograft. The disease involves an injury-response mechanism that incorrectly identifies small airways as foreign, caused by the upregulation of human leukocyte antigen (HLA) on respiratory epithelial cells. This misidentification prompts an immune response, potentially resulting in inflammation and lung damage [77]. Initial studies indicated that bronchiolar epithelial cells from former smokers exhibited both Class 1 and Class 2 HLAs [78]. However, antibody-mediated rejection is increasingly recognized as a significant cause of allograft injury, particularly in LTx [79,80]. Donor-specific antibodies, particularly those aimed at Class II HLAs like HLA-DQ2, significantly increase the risk of developing CLAD [81]. The rejection reaction is aided by ischemia, which is involved in scar formation following injury, and by disruption to the “watershed” microvasculature of terminal bronchioles [82]. A retrospective analysis of transbronchial lung biopsies supports an immunological etiology for BOS [83,84].

Understanding the pathogenesis of BOS is crucial to reduce the risk of BOS after LTx or HCT. Trigger events, such as T-cell-mediated acute cellular rejection, community-acquired respiratory viral infections, coronavirus infections, bacterial infections, and inflammation from gastric aspiration, can upregulate antigen presentation and damage small airways. Microvesicles containing self-antigens and autoantibodies against Type V collagen and K-α1 tubulin bolster autoimmunity to cryptic self-antigens released by these triggers. New therapies should integrate these concepts to avoid tissue damage and alter abnormal repair processes. A global response to air pollution and cigarette smoke exposure is also essential [4,85].

### 5.2. Aerosolised Liposomal Cyclosporine (L-CsA)

Cyclosporine in aerosol form provides targeted immunosuppressive treatment to the allograft, specifically targeting the immune activation site and minimizing overall exposure to the rest of the body. The US Food and Drug Administration has granted orphan drug status to a liposomal cyclosporine formulation for aerosolized delivery (Zambon, California, CA, USA), which is currently in phase III development [86]. The formulation demonstrated effective lung deposition when administered via nebulizer to lung transplant recipients. Another study highlighted that the inhalation of liposomal cyclosporine led to minor yet statistically significant improvements in lung function metrics, such as FEV1 [87]. However, it is important to note that this study was open-label and did not include a placebo control. The treatment was easily tolerated, and there was no significant increase in serious adverse events when using aerosolized L-CsA compared to standard care alone. Some of the adverse events reported included conjunctivitis, pharyngitis, and productive cough. Preliminary studies have shown that inhaled L-CsA is well tolerated and may help stabilize lung function in transplant recipients experiencing BOS [86]. The BOSTON-1 and BOSTON-2 trials specifically focus on treating BOS in single and double lung transplant patients, respectively [88]. The results from this ongoing research are anticipated to be available within the next 2–3 years, contributing valuable insights into post-transplant care and management [89].

### 5.3. JAK Inhibitors

JAK inhibitors are utilized for treating solid tumors and autoimmune disorders. Ruxolitinib is approved for steroid-refractory GVHD (SR-GVHD) in the USA and is the most researched agent for BOS following HCT or LTx [90,91]. The available data include small case series or studies involving adults and children with BOS. A phase II open-label study was conducted in the USA [92]. Abedin and colleagues investigated the efficacy of ruxolitinib in the treatment of SR-GVHD following allogeneic HCT. The study discovered that 42% of patients receiving ruxolitinib experienced an infectious event, with viral and bacterial occurrences being more common in acute SR-GVHD [93]. The study recommends that acute SR-GVHD patients be closely monitored for viral reactivation, continue with preventive antimicrobial treatments, and explore bacterial prophylaxis. The study questions whether bacterial prophylaxis is necessary for patients initiating ruxolitinib for acute SR-GVHD, especially those receiving high-dose steroids.

Itacitinib adipate, a selective JAK1 inhibitor developed by Incyte Corporation, is undergoing clinical trials aimed at treating BOS, particularly in patients following HCT [94]. This compound is currently in Phase 3 trials, focusing on its efficacy and safety in this specific patient population [95].

### 5.4. B-Cell-Directed Therapies

B-cells are essential in the development of chronic GVHD and CLAD, making treatments that inhibit B-cell activation valuable for BOS management following LTx or HCT. Rituximab, an antibody targeting CD20, may enhance lung function in patients with BOS following LTx, though data on its use in BOS after HCT are limited [96]. Research shows that rituximab may result in a modest decrease in the incidence and severity of cGVHD when administered preventively or preemptively after transplantation. The specific effects of rituximab on lung function in these patients are underexplored, indicating a need for targeted studies to address this knowledge gap [97]. Alemtuzumab, an anti-CD52 monoclonal antibody, reduces the 5-year risk of BOS after LTx, particularly benefiting early-stage BOS patients [98,99]. The use of alemtuzumab in this context is supported by its ability to effectively manage acute rejection episodes and CLAD, which are critical challenges in LTx [100]. However, this strategy could potentially increase the risk of anticancer and anti-infection control.

### 5.5. Antifibrotic Treatments

Pirfenidone and nintedanib, antifibrotic agents used in treating chronic fibrosing pulmonary conditions, have been shown to stabilize lung function and bridge to second lung transplants in some patients [101]. Clinical trials are ongoing in post-transplant patients with BOS, with the European Trial of Pirfenidone in BOS showing negative results [102]. Pirfenidone is also being studied in a small phase II trial for RAS patients. The randomized controlled study in four countries (UK, USA, Australia, and Canada) is exploring its potential role [103].

## 6. Conclusions

BOS management after LTx or HCT is similar, but systemic GVHD management is crucial. Overall, FAM therapy constitutes notable progress in the treatment of LTx and HCT recipients, effectively addressing BOS and enhancing patient outcomes and quality of life. Inhaled L-CsA can be administered safely and has the potential to stabilize or enhance lung functions in recipients of LTx and HCT with BOS, facilitating a reduction in systemic immunosuppression. The outcome showed limited systemic absorption in contrast to standard oral administration while also attaining elevated drug levels in lung tissue, as evidenced by the lung deposition results.

## Figures and Tables

**Figure 1 cancers-16-03715-f001:**
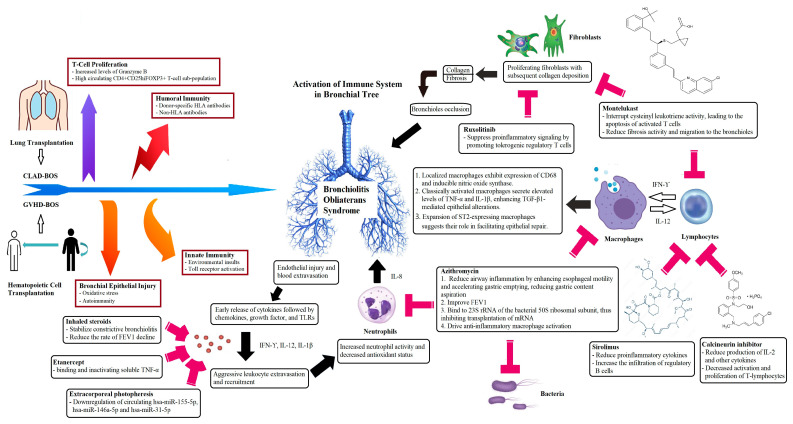
The pathogenesis of BOS is multifaceted, encompassing both immune responses and possible non-immune mechanisms that play a role in the progression of the syndrome. Abbreviations: BOS, bronchiolitis obliterans syndrome; CD, cluster of differentiation; CLAD, chronic lung allograft dysfunction; CysLTs: Cysteinyl leukotrienes; CysLT-R, Cysteinyl leukotrienes receptor; FEV1, forced expiratory volume in 1 s; FOXP3, Forkhead box protein P3; GVHD, graft-versus-host disease; HLA, human leukocyte antigen; has-miR, Homo sapiens microRNA; IFN, interferon; ILs, interleukins; TGF-β1, Transforming growth factor-beta1; TLR, Toll-like receptors.

**Table 1 cancers-16-03715-t001:** Pathogenesis, diagnosis, and treatment of BOS after lung transplantation or hematopoietic cell transplantation.

	CLAD-BOS [4]	GVHD-BOS [7]
Pathogenesis	Microbe-allograft-host interaction, causing allograft cells to release chemokines that recruit host leukocytes to the site, recognizing the airways as non-self	Chronic GVHD caused by central tolerance failure and B-cell and auto-antibody production
Diagnosis	Consistent drop in FEV1 of ≥20% from the baseline reference for over 3 weeks, after ruling out other potential causes	FEV1 < 75% projected with ≥10% decrease over <2 years should not surpass 75% with albuterol, and absolute decline should remain ≥10% over 2 years
	Obstructive pattern observed on spirometry (FEV1/FVC < 0.7)	FEV1/FVC ratio < 0.7, or anticipated fifth percentile
	No evidence of restriction	Absence of infection Another chronic GVHD manifestation
	No fibrosis detected in the pleura or lungs on CT	Two auxiliary features of BOS: (1) air trapping on expiratory CT, minor airway thickness or bronchiectasis on high-resolution chest CT, or (2) PFT
Treatment	First-line: CNI switch azithromycin and montelukast	First-line: Systemic steroids, FAM
	Second-line: ATG, IVIG, rituximab, JAK inhibitors	Second-line: ECP, JAK inhibitor
	Ultimate therapy option: Re-transplantation	Ultimate therapy option: Lung transplanatation

Abbreviation: ATG, anti-thymocyte globulin; BOS, bronchiolitis obliterans syndrome; CLAD, chronic lung allograft dysfunction; CT, computed tomography; ECP, extracorporeal photopheresis; FAM, fluticasone, azithromycin, and montelukast; FEV1, forced expiratory volume in 1 s; FVC, forced vital capacity; GVHD, graft-versus-host disease; JAK, Janus kinases; PFT, pulmonary function test.

**Table 2 cancers-16-03715-t002:** Issues in the Diagnosis and Management of Post-HCT BOS and Potential Solutions.

Identified Issue	Possible Solution
PFT reference equation is outdated	Switch to global lung initiative global reference equations
Spirometry as a sole method	Criteria established for individuals unable to undergo spirometry Incorporation of MBW
Requirement of FEV1 < 75%	Evaluate decline compared to pre-HCT baseline
Ignoring the possibility that infection and BOS can coexist	Emphasize the critical need for diagnosing infections in immunocompromised patients while recognizing the possibility of concurrent infection and BOS
Failure to account for variations in FEV1/FVC	Incorporate mid-expiratory flows, PRISm, and MBW-based indices.
Enhance sensitivity and facilitate early diagnosis	Include additional supporting features (e.g., MBW, mid-expiratory flows, PRISm). Include an at-risk stage that necessitates evaluation and closer follow-up.
Exclusion of pulmonary complications aside from BOS	Create broader criteria for all post-HCT pulmonary complications
Potential adverse effects associated with ruxolitinib therapy	Be mindful of common issues such as anemia, thrombocytopenia, and the risk of infection.

Abbreviation: BOS, bronchiolitis obliterans syndrome; FEV1, forced expiratory volume in 1 s; FVC, forced vital capacity; HCT, hematopoietic cell transplant; MBW, multiple breath washout; PRISm, preserved ratio impaired spirometry; PFT, pulmonary function test.

## Data Availability

Not applicable.

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
