# Peer review of "Time to Rethink Bronchiolitis Obliterans Syndrome Following Lung or Hematopoietic Cell Transplantation in Pediatric Patients"

_cancers, 2024, doi:10.3390/cancers16213715_

Round 1

Reviewer 1 Report

Comments and Suggestions for Authors

This is a comprehensive review of lung allograft rejection and brochiolitis obliterans following HCT.These disorders have several similarities and it is an excllent idea to combine both ecperiences.In this way, the article will attract readers who are thoracic surgeons performing lungtransplants as well as colleagues performingHCT.Furthermore,lungtransplantation may be a treatment for obstructive broncheolitis following HCT.

In the  introduction it is mentioned that this review is regarding pediatric patients.
Maybe this should also be stated in the title.

The article is well written and easy to read.When reading a review you want the whole field covered with a more or less complete lists of references in the field.This reviewer requests more references.

Author Response

This is a comprehensive review of lung allograft rejection and bronchiolitis obliterans following HCT. These disorders have several similarities and it is an excellent idea to combine both experiences. In this way, the article will attract readers who are thoracic surgeons performing lung transplants as well as colleagues performing. Furthermore, lung transplantation may be a treatment for obstructive bronchitis following HCT.

In the introduction, it is mentioned that this review is regarding pediatric patients.

Maybe this should also be stated in the title.

The article is well-written and easy to read. When reading a review you want the whole field covered with a more or less complete list of references in the field. This reviewer requests more references.        

Response:

Your feedback is greatly valued. The title has been updated to “Time to Rethink Bronchiolitis Obliterans Syndrome Following Lung or Hematopoietic Cell Transplantation in Pediatric Patients” in accordance with your comments.

Reviewer 2 Report

Comments and Suggestions for Authors

This paper addresses the diagnosis, immunological basis of onset, and treatment of bronchiolitis obliterans syndrome (BOS), which develops following lung or hematopoietic cell transplantation. It emphasizes the importance of early diagnosis and effective management of BOS, alongside introducing new treatment approaches. While the content is succinctly summarized, further elaboration is needed in the following points.

 1.    Table 1 compares the diagnostic criteria for BOS after lung transplantation and after hematopoietic cell transplantation. However, creating similar tables for the pathogenesis and treatment methods would allow for a clearer understanding of the similarities and differences between each type of BOS. 

2.    The immunological mechanisms underlying BOS pathogenesis are not sufficiently detailed. Further explanation is necessary, with a focus on the content presented in Figure 1. 

3.    While various treatment examples are provided, the treatment algorithm covering the progression from initial management to chronic phase management of BOS is not clearly outlined. Specific treatment recommendations for each stage need to be described more explicitly. 

4.    The paper attributes delays in BOS diagnosis to inadequate lung function monitoring but lacks a detailed discussion of specific countermeasures. It would be beneficial for the paper to include more specific recommendations, such as whether frequent lung function tests are advocated or if new diagnostic monitoring methods should be developed. 

5.    Figure 1 provides a comprehensive overview of topics ranging from pathogenesis to treatment methods. However, the content in the figure is not adequately explained in the main text or figure legend. It is essential to revise the figure legend and main text to ensure that the information in the figure aligns clearly with the text and figure legend.

Author Response

Reviewer 2

This paper addresses the diagnosis, immunological basis of onset, and treatment of bronchiolitis obliterans syndrome (BOS), which develops following lung or hematopoietic cell transplantation. It emphasizes the importance of early diagnosis and effective management of BOS, alongside introducing new treatment approaches. While the content is succinctly summarized, further elaboration is needed on the following points.

  1. Table 1 compares the diagnostic criteria for BOS after lung transplantation and after hematopoietic cell transplantation. However, creating similar tables for the pathogenesis and treatment methods would allow for a clearer understanding of the similarities and differences between each type of BOS.

Response:

Following your recommendation, the pathogenesis and treatment methods have been incorporated into the context of Table 1.

  1. The immunological mechanisms underlying BOS pathogenesis are not sufficiently detailed. Further explanation is necessary, with a focus on the content presented in Figure 1.

Response:

The pathogenesis of BOS related to lung transplantation and HCT has been incorporated into Table 1 as detailed below:

  1. While various treatment examples are provided, the treatment algorithm covering the progression from initial management to chronic phase management of BOS is not clearly outlined. Specific treatment recommendations for each stage need to be described more explicitly.

Response:

Following your advice, the first-line, second-line, and ultimate therapy options have been included in Table 1 for a clearer description.

  1. The paper attributes delays in BOS diagnosis to inadequate lung function monitoring but lacks a detailed discussion of specific countermeasures. It would be beneficial for the paper to include more specific recommendations, such as whether frequent lung function tests are advocated or if new diagnostic monitoring methods should be developed.

Response:

Thank you for your esteemed suggestion. We have incorporated a paragraph in the Diagnosis Section, which now reads “Recent research emphasizes the significance of early diagnosis in enhancing patient outcomes. New CT strain metrics allow for earlier detection of BOS, even before significant declines in traditional measures such as FEV1 [37]. This advancement is vital as BOS can appear as early as three months after transplantation, resulting in progressive obstructive lung disease that particularly affects peripheral airways [38].”

  1. Figure 1 provides a comprehensive overview of topics ranging from pathogenesis to treatment methods. However, the content of the figure is not adequately explained in the main text or figure legend. It is essential to revise the figure legend and main text to ensure that the information in the figure aligns clearly with the text and figure legend.

Response:

We made our best attempt to revise the figure. The content of Figure 1 has been revised and expanded to ensure that the information aligns clearly with both the text and the figure legend.

Reviewer 3 Report

Comments and Suggestions for Authors

The Authors describe post-transplant bronchiolitis obliterans with an accurate and interesting review.

The data reported are very interesting and recently relevant as the complication that occurs can be a serious post-transplant problem.

The description of the review allows clinicians, doctors and thoracic surgeons to improve their knowledge in this severe complication if not treated early.

The work is well written, the English is clear and the data reported in the various sessions shed light on this important complication.

The tables are also clear and make the text easier to read.

The conclusions are relevant to the aim of the work.

Author Response

Your feedback is valued and appreciated. However, certain adjustments were made to the manuscript's content, including the figure, to enhance clarity regarding this enigmatic syndrome for the reader.  

Round 2

Reviewer 2 Report

Comments and Suggestions for Authors

Thank you for addressing the comments and suggestions made in the previous review. I have carefully reviewed the revised manuscript and am pleased to note that the authors have made appropriate adjustments to the points raised. The changes effectively clarify the key aspects of the study, and the additional details contribute to a better understanding of the research context and results.